# Cost-Effective Healthcare in Rehabilitation: Physiotherapy for Total Endoprosthesis Surgeries from Prehabilitation to Function Restoration

**DOI:** 10.3390/ijerph192215067

**Published:** 2022-11-16

**Authors:** Tünde Szilágyiné Lakatos, Balázs Lukács, Ilona Veres-Balajti

**Affiliations:** 1Clinical Center Gyula Kenezy Campus Clinic of Medical Rehabilitation and Physical Medicine, University of Debrecen, 4031 Debrecen, Hungary; 2Department of Physiotherapy, Faculty of Health Sciences, University of Debrecen, 4028 Debrecen, Hungary

**Keywords:** preoperative, endoprosthesis, postoperative, rehabilitation, guideline

## Abstract

Knee and hip joint replacements for the elderly are increasingly placing a burden on healthcare. Our aim was to verify the efficiency of the prehabilitation program among patients with knee arthroplasty (TKA) and hip arthroplasty (THA), taking into account the length and cost of postoperative rehabilitation and the restoration of function. We introduced a two-week preoperative physiotherapy program for patients awaiting knee and hip replacement surgery. We measured the duration and costs of the hospital stays, the active and passive range of motion of the hip and knee joints, and the quality of life. In the study, 99 patients participated (31 male, 68 female), with a mean age of 69.44 ± 9.69 years. We showed that, as a result of the prehabilitation program, the length of postoperative hospital stay decreased (THA: median 31.5 (IQR 26.5–32.5) vs. median 28 (IQR 21–28.5), TKA: median 36.5 (IQR 28–42) vs. median 29 (IQR 26–32.5)), and the patients’ quality of life showed a significant improvement (TKA: median 30.5 (IQR 30–35) vs. median 35 (IQR 33–35), THA: median 25 (IQR 25–30) vs. median 33 (IQR 31.5–35)). The flexion movements were significantly improved through prehabilitation in both groups. Based on our positive results, we recommend the introduction of prehabilitation into TKA- and THA-related care.

## 1. Introduction

Osteoarthrosis is a chronic disease affecting the joints. It affects more than one third of the adult population worldwide, and its prevalence increases with age [1]. Osteoarthrosis is one of the main causes of disability and is closely related to the physical activity of the patient. An aging population and the rising prevalence of osteoarthrosis (OA) have generated a higher demand for prosthetic procedures and the subsequent complex rehabilitation services. This increased demand places an additional burden on healthcare providers [2]. Once OA has developed, there are multiple treatment options available, the most prominent of which are regular exercise and special physiotherapy. In addition to physiotherapy, patients can also undergo medication therapies, the most widespread of these being non-steroidal anti-inflammatory drugs (NSAID). In patients who cannot take oral NSAID, or who fail to achieve adequate therapeutic effects, intra-articular hyaluronic acid injections or intra-articular corticosteroid injection can be used [3]. If the pain is already so great that it hinders the patient’s movement and self-sufficiency, surgery will take place. To help the patient to regain the highest degree of self-sufficiency, rehabilitation after surgery is of paramount importance. As explained by Howard-Wilsher et al. [4], there is a need to develop and disseminate rehabilitation services to restore health as soon as possible. Well-organized, structured rehabilitation interventions can lead to cost savings and also cost reductions in other healthcare services [5]. Based on our experience, according to the Hungarian practice, patients participate in a rehabilitation program lasting for an average of 6–8 weeks after surgery. During this time, the range of joint movement and the muscle strength required for walking can be improved, and patients can learn how to use different walking aids if required. They are also taught the movements that are contraindicated for the given postoperative condition. These serve as a compass for their activities after being released [6]. In many countries, the practice of patients’ participation in so-called “prehabilitation” before endoprosthesis surgery is becoming more common. The aim of prehabilitation is to reduce the recovery time and hospital stay. The theory of prehabilitation is based on the fact that patients with higher functional abilities and endurance are better able to tolerate surgical interventions. Research has shown that patients with higher fitness levels have a reduced rate of postoperative complications and show better functional and psychosocial results. Numerous studies have been conducted to evaluate the effectiveness of preoperative exercise among patients awaiting knee or hip replacement surgery [7]. Jungae An et al. examined the effect of the preoperative telerehabilitation (PT) program on the range of motion (ROM) of the knee among patients who underwent knee endoprosthesis, as well as the effect on the functional state of the patients. The intervention took 3 weeks, with patients undergoing the movement therapy in their homes, assisted by physiotherapists. The results showed that the program improved the muscle strength, ROM, and functional results before surgery, which in turn contributed to better functional restoration after arthroplasty [8]. Oosting E. et al. investigated the effectiveness of exercise programs preceding hip prosthesis surgery in terms of the functional movement and gait capacity. Their postoperative results showed a significant improvement in the 6 min walk test and the chair rise time test compared with the control group [9]. Majid et al. compiled a summary of research published between 2003 and 2013 on the effectiveness of patient education among patients awaiting orthopedic surgery. They examined the length of hospital stay, patient satisfaction and level of pain, cost of care, changes in functional abilities, knowledge, anxiety, and changes in quality of life. The study found that preoperative patient education has a distinctly positive effect on the success of the rehabilitation program along the lines described above [10]. With the help of Medline, PubMed, Embase, CENTRAL, CINAHL, and Ageline, Rajrishi Sharma and colleagues examined articles investigating the importance of prehabilitation for patients undergoing unilateral knee replacement surgery [11]. The aim of the study by Pascale Gränicher et al. was to assess the effect of preoperative physiotherapy on functional, subjective, and socioeconomic parameters after total knee surgery (TKA) [12]. Jones et al., as well as Huang et al., found a reduction in the cost and length of hospital stay after applying prehabilitation [2,13]. Huifen Chen et al. also investigated the ways in which prehabilitation practices affect the hospital stay in the post-rehabilitation period and improvements in the knee movement in degrees (ROM) [14]. Robert Topp and colleagues also support the conclusion that prehabilitation is important [15]. Swank et al. examined of 4- to 8-week function-enhancing prehabilitation program [16].

The aim of our study was to demonstrate that, as a result of prehabilitation, the length of postoperative rehabilitation is reduced after a complete hip or knee replacement, thereby reducing the postoperative hospital costs. A further aim of our research was to examine the efficiency of prehabilitation in terms of the increase in the motion range of the joints and walking distance, as well as changes in quality of life.

## 2. Methods

A two-week preoperative physiotherapy program was introduced for patients awaiting knee and hip replacement surgery at the Rehabilitation Department of Gyula Kenézy University Hospital (DE KEK), University of Debrecen. To this end, a new patient pathway was developed by the hospital management and the heads of the DE KEK Rehabilitation Department and the Department of Traumatology and Hand Surgery in May 2018.

The process begins with an appointment with the patient, on the basis of which the specialist schedules the date of the surgery. The patient is then given a recommendation by the physician who is scheduled to perform the surgery advising them to participate in a prehabilitation program, which is designed to boost the patient’s postoperative recovery. The prehabilitation program takes place in one of the departments of the Rehabilitation Department in the 6th and 5th weeks before surgery. If the patient agrees to participate, a consultation with the Rehabilitation Department will determine whether the patient will undertake the preparatory program on an outpatient or inpatient basis. If the patient prefers inpatient care, they may opt for only a hospital bed and one daily meal for the duration of the treatment, and the full hospital hotel service will not be employed. This form of care is called a day hospital. Inpatient care is only recommended if the patient is unable to make their daily visit to the hospital. Whichever option the patient chooses, they receive the same high-quality physiotherapy care in every case. The functional recovery of patients after surgery is also performed in the Rehabilitation Department. Patients receive individual daily exercise, supplemented with physiotherapy if necessary.

The target group included patients scheduled for knee or hip endoprosthesis surgery in the Department of Traumatology and Hand Surgery, DE KEK. The program was implemented in May 2018 at the hospital. The patients involved in the program were divided into two groups. The intervention group consisted of those who consented to participate in the preoperative prehabilitation program. Patients who did not undertake the program were included in the control group. The estimated sample was calculated using the “sampsi” command of Intercooled Stata v13. The type I error was set to 5%, and the statistical power was set to 90%. The refusal rate was also taken into consideration. The calculation was based on a relevant article [17]. The size of our study sample was determined according to the hospital’s surgery capacity and considering previous publications, where the efficiency of the prehabilitation and postoperative treatments was examined among patients with hip and knee replacement [17,18].

The exclusion criteria for both groups were:-The presence of any comorbidity meaning that the general postoperative functional rehabilitation program was inapplicable, e.g., previous limb amputation, paralysis of central or peripheral origin, or chronic arthritis not affecting the operated joint.-A lack or weakness of cooperation with the rehabilitation team and the objectives we evaluated based on a personal interview.

The inclusion criteria for the intervention groups were:

-An ability to participate in the prehabilitation program on a daily basis and carry out the gymnastics program established by the team for 30 min a day.-Willingness to cooperate with the professionals involved in the prehabilitation program and accept and follow instructions.

During their preoperative education, the patients see a doctor, a physiotherapist, and, if the team deems it necessary, a psychologist. The main part of the complex physiotherapy program for the target group is exercise. The exercise focuses on improving the functional status and breaking down the muscle balance in the lower limbs using various physiotherapy techniques. The tasks are performed by the patients in a gradual manner according to their individual abilities. The patients take part in a 30 min individual exercise program every day for two weeks.

In addition to the exercise program, electrotherapy and massage treatments were also part of the intervention, aiming to reduce pain and provide targeted muscle relaxation.

The program included patient education regarding the proper use of the tools used in the postoperative period. Doctors and physiotherapists also created strategies for the patients’ altered way of life that the patients took home with them after being discharged from hospital. During surgery, the patients underwent total hip and knee replacements. The fixation was achieved either with or without cement. In both types of surgery, the surgical procedure took place in agreement with the professional protocol of The Ministry of Health [6,18]. The recruitment of the patients is shown in Figure 1.

The post-surgery care of patients was performed in the Traumatology Department, where patients, after surgery, had access to individual physical therapy. This was when those who did not take part in the prehabilitation program became acquainted with the use of the proper aids and forbidden movements.

On the 8th to 10th days after surgery, following suture removal, the patients were transferred to the rehabilitation ward for a medical consultation and physiotherapy assessment on the same day, with post-operative rehabilitation starting on the first day. For patients with hip replacements, the strengthening of the hip flexors and abductors was performed in the supine and lateral position using isometric and isotonic exercises, paying close attention to the gradual load. This was followed by increasing the motion range of the hip joint using guided active and active exercises. For patients who underwent knee replacement surgery, the main aim was to strengthen the knee flexors and extensors and to increase the motion range. To increase the motion range, we used CPM, while to strengthen the muscles, we also used electrotherapy equipment. A further important element of the postoperative program was ergotherapy, designed to improve the functional movement. The novel patient pathway is shown in Figure 2.

### 2.1. Data Collection

#### 2.1.1. Questionnaire

Oxford Hip and Knee Score

The changes in the patients’ quality of life were evaluated using a questionnaire method. The internationally used and validated Oxford Knee and Hip Score quality of life scale was used for the data collection. The intervention group completed the questionnaires after both the preoperative and postoperative rehabilitation programs, while the control group did so only after the postoperative program. Completed by the patients, the Oxford Hip and Knee Score contains 12 questions about pain and functional ability over the previous four weeks. The questions are scored with numbers between 1 and 5, with 5 meaning normal function and 1 meaning severe difficulty. The scores are summed to give a total score between 12 and 60. A lower score indicates a more severe condition [19,20].

#### 2.1.2. Physical Examinations

To describe the patients’ functional status, we measured the active and passive ranges of motion of the hip (flexion, extension, abduction) and knee (flexion, extension) joints and walking distance without rest.

(1)Range of motion tests

The active and passive ranges of motion of the joints were measured using a goniometer [21]. Measurements were performed at the beginning and end of both the preoperative and postoperative exercise programs in the intervention group and at the beginning and end of postoperative rehabilitation in the control group. The measurements were conducted three times and averaged.

(a)Hip
FlexionTo measure the hip joint flexion, we had the patient lie in the supine position, with the lower extremity under examination stretched out and the other extremity bent. The inflexion point of the goniometer was placed on the trochanter major. The stable arm of the goniometer was parallel to the body, and its mobile arm was parallel to the femur. The patient was asked to pull their lower extremity to their abdomen, and following their movement with the mobile arm of the goniometer, we measured the flexion in the hip, whose physiological value was 110–130° [22].ExtensionStart position: The patient is prone. The hips and knees are in the neutral position. Stabilization: The pelvis is stabilized by the therapist’s hand. Goniometer arm: The axis is placed over the greater trochanter of the femur. Stationary arm: Parallel to the midaxillary line of the trunk. Movable arm: Parallel to the longitudinal axis of the femur, pointing toward the lateral epicondyle. End position: The patient’s knee is maintained in extension. The hip is extended to the limit of motion at 30° [22].AbductionStart position: the patient is supine, with the lower extremities in the anatomical position. One must ensure that the pelvis is level. Goniometer axis: The axis placed over the ASIS on the side being measured. Stationary Arm: Along the line between the two ASISs. Movable arm: Parallel to the longitudinal axis of the femur. In the start position described, the goniometer indicates 90°. This recorded as 0°. End position: The hip is abducted to the limit of motion at 45° [22].

(b)Knee
FlexionWhen measuring the active flexion of the knee joint, we had the patient lie in the supine position with the lower extremity under examination extended and the other pulled up. The inflexion point of the goniometer was placed on the fibular head, the stable arm was parallel to the femur, and the mobile arm was pointing towards the medial malleolus. The patient was asked to pull their heel up to their buttocks as far as they could while the examiner followed the movement using the mobile arm of the goniometer. The physiological value of the passive flexion of the knee joint was 130–140° [22].ExtensionWhen measuring the active extension of the knee joint, we had the patient lie in the supine position with the lower extremity under examination extended. The inflexion point of the goniometer was placed on the fibular head, the stable arm was parallel to the femur, and the mobile arm was pointing towards the medial malleolus. The normal value of the extension of the knee joint was 0° [22].

(2)Measuring walking distance

The patient was asked to walk on a flat surface without rest using an aid appropriate to their physical condition until the occurrence of pain. The distance covered was proportionate to the functional condition of the patient. In the walking distance assessment, the patients were divided into three categories based on their performance. The first category was 0–50 m, the second category was 51–200 m, and the third category was more than 200 m.

#### 2.1.3. Intervention

During the prehabilitation program for hip and knee joints, the daily exercise lasted for 30 min. It consisted of 3 parts:

Introduction: The patients were in a relaxed state, resulting in reduced muscle tension caused by pain. We increased the circulation in the lower extremities using leg presses and ankle circles. We asked the patients to perform breathing exercises, mobilizing the upper extremities and the chest. The aim of these exercises was to prepare the cardio-vascular system for a physical load to support tissue oxygenation and to achieve psychological stress relief.

I.Increasing the range of motion of the affected hip or knee joint:

Hip joint:
At first, the improvement of the flexion was performed by patients in the supine position through passive/assisted active movements guided by a physiotherapist, starting with the short lever arm followed later by active movements, including short as well as long lever arms. In the lateral position, we had patients practice active flexion movements only.Abduction and adduction movements were carried out by patients in the supine and lateral positions. At first, the abduction was performed against gravity, and then, to increase the resistance, smaller ankle weights (0.5 kg) or flexible rubber bands were used to make the agonist–antagonist reflex muscle relief effect more effective.Extension movements were performed in the supine position using isometric exercises, mainly through the activation of the gluteal muscles, and in the lateral position through active movements. Finally, for a short period of time (5–10 min max), movements were performed with the patients positioned in the prone position.During the practice of plane movements, great emphasis was placed on the maintenance of the corrected rotation middle position.

2.Knee joint:
During the increase in the range of flexion–extension movement, all three positions were used (supine, prone, and lateral), using only active relaxation and stretching techniques (post-isometric relaxation; reciprocal innervation; contract–relax PNF stretching). In the case of, extension particular attention was paid to practicing the movement while maintaining a full range of motion.

II.Increasing the muscle strength:

Here, the emphasis was on the strengthening of the extensor and abductor muscles, as well as the quadriceps. In the case of the hip joint, our aim was to correct the muscle imbalance caused by typical flexion and abduction contracture. To this end, the most frequently used positions were the classical horizontal positions (supine, prone, lateral, and all fours) and the alternative variants of these, e.g., when the patient is lying on the therapy bed in the prone position at hip height and the two lower extremities are positioned towards the floor, hanging from the end of the bed. In this position, concentric and eccentric movements of the hip extensors can easily be performed, and the effectiveness of the training can be further increased using less resistance. In addition, if the patients’ general condition allows for it, they can perform muscle strengthening exercises using wall bars. During these exercises, the concentric and eccentric strengthening of the abductors performed in a corrected position is mainly practiced using different levels of resistance.

A further aim of our study was to strengthen the quadriceps muscles, which are characterized by the considerable weakening of both the hip and the knee joints in degenerative diseases. Typically, in our study, this strengthening was carried out with patients in the supine position, targeting the affected leg, with an extended hip joint in starting position, working with short and long lever arms. We increased the load gradually by increasing the number of repetitions of the exercises, using gravity only or, in some cases, weaker rubber bands for resistance. In order to avoid para-coordination or compensatory movements, the pelvis was always stabilized by the positioning of the other lower leg, or, if necessary, manual fixing by the physiotherapist.

The intensity and strength of the exercises, as well as the number of repetitions, were determined in every case based on the current condition of the patient. We ensured that we did not cause or increase pain during the movements.

The tools used during the exercise program helped to maintain the patients’ interest and meant that the exercises were more varied. Soft balls, rubber bands of different strengths, Dynair cushions, and fit balls were used.

III.Cooldown:

The last stage of the exercise program featured a gradual load decrease, as well as relaxation exercises and active stretching and breathing exercises to restore circulation.

During the prehabilitation program, the members of the intervention group also received electrotherapy and therapist massages in addition to the exercises. The electrotherapy treatments were meant to reduce pain in the given area, improve circulation, and activate and strengthen the muscles around the joints. To this end, interferential therapy, the application of selective stimulation, and electromagnetic therapies were also incorporated into the program.

By using classical therapist massage, we intended to increase the range of motion of the joint, improving circulation and relaxing painful adhesions.

### 2.2. Statistical Analysis

The statistical analysis was performed using Microsoft Office Excel and SPSS software. Most of the data did not follow a normal distribution (which was tested using the Shapiro–Wilk normality test), which is why the statistical analysis of the control and intervention groups (after surgery vs. leaving hospital) was performed with the Wilcoxon signed rank test, and the comparison between the control and intervention groups was conducted using the Mann–Whitney U test. The gender distribution was tested using Fisher’s exact tests. Results were considered significant at *p* < 0.05.

### 2.3. Ethical Approval

The study was approved by the Research Ethics Committee of the Kenézy Gyula Hospital of the University of Debrecen (É/2020/01/23), and the participants gave informed consent to the collection and processing of their data.

## 3. Results

### 3.1. Demographical Data

A total of 99 people participated in the study. We examined 57 patients for hip prosthesis surgery, 27 of whom were in the intervention group (exposed), including 16 women and 11 men, with a mean age 66.23 years. The control group consisted of 30 people (unexposed). The mean age of the 22 women and 8 men was 71.63 years. For the implantation of the knee prosthesis, 42 patients were included in our study. The intervention group consisted of 14 people, including 11 women 3 men with an average age of 71.5 years. There were 28 patients in the control group. There were 19 women and 9 men, with a mean age of 69.86 years.

In terms of gender, there was no significant difference between the control and intervention groups in the TKA group, but a significant difference was found in the THA group, according to which the men in the intervention group were significantly younger than those in the control group. There was no difference in the proportion of the genders in the control and intervention groups among either the TKA (*p* = 0.719) or THA group (*p* = 0.278).

The demographic data are shown in Table 1.

### 3.2. Days Spent in the Hospital and Healthcare Costs

Among the patients with hip prosthesis implantations, the postoperative hospital stay was significantly (*p* < 0.01) shorter in the intervention group than in the control group (median 31.5 (IQR 26.5–32.5) *n* = 27 vs. median 28 (IQR 21–28.5), *n* = 30). In the intervention group, the number of inpatient days after surgery decreased, resulting in a reduction in the postoperative costs of approximately 26%. Another favorable outcome of preoperative preparation was that five patients did not require postoperative rehabilitation in a hospital setting. They used the in-home physiotherapy service, in their family environment, further reducing the cost of hospital care.

Among the patients with knee prosthesis implantations, the postoperative hospital stay was significantly (*p* < 0.05) shorter in the intervention group than in the control group (median 36.5 (IQR 28–42) *n* = 14 vs. median 29 (IQR 26–32.5) *n* = 28)). Similar to the hip prosthesis patients, a reduction in the postoperative costs of approximately 16% was observed among those who underwent knee surgery due to their shorter hospitalization.

### 3.3. Range of Motion of the Hip and Knee Joints

For both the hip and knee joint movements, the intervention group achieved better results by the end of the postoperative period. When determining the flexion in the hip surgery patients, we measured a significantly better value in the intervention group after the operation compared to the control group (median 75° (IQR 70–80°) *n* = 27 vs. median 60° (IQR 50–70°), *n* = 30). Among the knee surgery patients, the flexion showed a significant difference compared to the controls only at the endpoint of the rehabilitation (median 100° (IQR 95–100°), *n* = 14 vs. median 90° (IQR 90–90°), *n* = 28). The extension values of both the hip and knee surgery patients tended to be better in the intervention groups compared to the control patients, but they failed to reach statistical significance. Regarding the hip abduction values, patients in the intervention group showed better results. However, neither at the beginning nor at the end of the rehabilitation were we able to show a significant difference between the intervention and the control groups. The results are shown in Table 2.

### 3.4. Walking Distance

In the case of hip replacement surgery, 84% members of the intervention group were able to walk less than 50 m and 16% were able to walk more than 50 m at the beginning of postoperative rehabilitation. In comparison, however, only 7% of the members of the control group were able to walk more than 50 m at the beginning of their rehabilitation. At the end of rehabilitation, 68% of the intervention group belonged to the second (51–200 m) and third (>200 m) categories, and 14 patients could walk between 50 m and 200 m, while 3 patients were able to walk more than 200 m without rest. In the control group, the proportion of patients falling into the second and third categories was only 50%.

In the case of knee endoprosthesis surgery, at the start of the rehabilitation program, only 7% of the patients could walk more than 50 m in the control group, while in the intervention group, all the patients belonged to the first category. At the end of rehabilitation, 61% and 57% of the control and intervention patients could walk more than 50 m without rest, respectively.

### 3.5. Oxford Hip and Knee Score

Among the patients who received hip endoprosthesis surgery, the Oxford Hip and Knee Score of the intervention group was significantly (*p* < 0.001) higher than that of the control group (median 33 (IQR 31.5–35), *n* = 27 vs. median 25 (IQR 25–30), *n* = 30). Among the knee endoprosthesis surgery patients, the Oxford Hip and Knee Score was also significantly (*p* < 0.05) higher in the intervention group than in the control group (median 35 (IQR 33–35), *n* = 14 vs. median 30.5 (IQR 30–35), *n* = 28). The results are shown in Table 3.

## 4. Discussion

Our research investigated the efficiency of a novel and special patient pathway in the course of which a patient receives a recommendation for participation in prehabilitation from the physician who will operate on them. During the prehabilitation program, the patient meets the members of the intervention team, including the specialists with whom they will work together with in the post-rehabilitation period. This option increases the patient’s trust in the successful outcome of the surgery.

Our results showed that the patients in the prehabilitation program were able to leave the hospital 7.5 days earlier, on average, in the postoperative rehabilitation process compared to patients in the control group. In addition, the results of the range of motion measurements showed that the intervention group was able to achieve better functional abilities during postoperative rehabilitation despite their shorter hospital stays. These results were in accordance with those reported by Oosting E. et al., who found a significant difference between prehabilitation and control patients undergoing hip endoprosthesis surgery in the timed up and go test (2.9 s; 95% confidence interval (CI), 0.9–6.6) and the 6 min walk test (41 m; 95% CI, 8–74) [8]. In their systematic review and meta-analysis, Moyer et al. also found that the preoperative exercise and education of patients undergoing total hip and knee arthroplasty significantly improved the postoperative function in both patient groups (THA, TKA), with similar improvements. Furthermore, in their review, the length of hospital stay was significantly shorter after TKA and THA [23]. Dlott et al. also demonstrated that elective primary total hip and knee arthroplasty patients’ education, delivered by nurses before the elective total hip and knee arthroplasty, reduced the average length of hospital stay and number of postoperative emergency department (ED) visits, and subsequently increased the use of telerehabilitation [24].

In their review, Widmer et al. compared the two forms of prehabilitation, exercise therapy and patient education, among patients with total hip arthroplasty (THA) [25]. They concluded that, in the postoperative period, the exercise-based prehabilitation patients had better results in the functional tests compared with patients who received no prehabilitation. In contrast, patient education alone did not significantly improve the patients’ results [25,26].

Several previous studies and summary reports have examined the effectiveness of prehabilitation among patients with hip and knee replacements [26,27,28]. However, the results are ambiguous, a fact possibly explained by differences in the types of interventions used during prehabilitation and the lengths of the prehabilitation periods. The importance of the time factor was also highlighted by Widmer et al. The effectiveness of prehabilitation shows a close relationship with the length of the intervention and the intensity of the therapies. In other words, the effectiveness of prehabilitation shows dose dependence in the case of both outpatients and inpatients [25]. The question arises as to whether the effectiveness of prehabilitation is affected by socio-demographic factors (e.g., age or gender). However, to date, the role of these factors has not been confirmed in the case of patients with hip replacements [25].

In our study, the data based on the quality of life scales for the intervention group showed a very significant difference associated with hip prosthesis surgeries, and the patients receiving knee prosthesis also reported a significantly better quality of life. The quality of life scale data showed highly significant differences in the intervention group associated with hip replacement surgery, and patients who underwent knee replacement also reported a significantly better quality of life.

Due to the shorter hospital stays, the cost of hospital care was reduced, resulting in financial savings for the healthcare system. In our research, the postoperative costs of hip and knee endoprosthesis implantations decreased by 26% and 16%, respectively. Our results are comparable to those of previous studies, such as those of Jones et al. and Huang et al., who reported that patients in the study spent fewer days in hospital and thus had more favorable hospital costs (mean: 7.12 days, *p* = 0.027, and mean: 123.73, which is USD 427.4 or EUR 302.1, *p* = 0.001) compared to the control group [2,13].

The efficiency of our prehabilitation program was demonstrated by the fact that three of our patients who underwent hip replacement surgery did not need postoperative rehabilitation in hospital at all due to their favorable functional condition. Butler et al. and Konnyu et al. demonstrated that THA patients who undergo prehabilitation require a shorter rehabilitation, as they reach adequate functionality and can leave the hospital earlier [27,28].

The results obtained in our study support the usefulness of the new patient pathway, and our results are, in many respects, consistent with those of previous reports. In their studies, Rajrishi Sharma et al. found a significant result in terms of the hospital stay in favor of prehabilitation [11]. The results of studies conducted by Pascale Gränicher et al. also suggest that preoperative therapy influences the patients’ movements in a positive direction [12]. In the conclusion of their article, Plenge et al. described the importance of developing a perioperative multidisciplinary patient care protocol for hip and knee arthroplasty patients based on the results of preoperative interventions [29].

## 5. Limitations

By international standards, the patients whom we examined stayed longer in the Rehabilitation Ward than their counterparts in other European countries. This is due to several reasons. In Hungary, home care is poorly supported, as a result of which care providers cannot employ sufficient numbers of rehabilitation professionals. This results in long waiting lists which, in turn, lead to delays in the start of therapy. Thus, patients find it safer to spend the rehabilitation period in hospital, enabling them to use the services of a professional staff (nurses, physiotherapists, and physicians) 24 h a day. This is something that is enabled and financed by the health care system.

We analyzed the total hospital cost of patient care. The total cost per patient for prehabilitation increased significantly by the end of the program, because the majority of the patients chose to stay in hospital for the prehabilitation program. A solution to this problem could be to have patients attend prehabilitation in a day hospital or outpatient setting, with the help of family or support services who could facilitate their day-to-day visits.

The low number of cases also caused difficulties. The number of patients in the intervention group was limited by several factors, as follows. 1. Many members of the target group could not be included in the study due to the exclusion criteria. 2. The surgeons were worried that the prehabilitation that took place in the hospital could lead to an increased risk of nosocomial infections, which, in turn, could result in the postponement of the planned surgery. This is why, in many cases, they did not recommend the patient’s inclusion in the program. 3. In our study, the average age of the patients undergoing replacement surgery was 69.44 ± 9.68. These patients were particularly affected by the experience of being away from their families. Hence, they turned down the option of postoperative hospital stay, and they could not arrange outpatient care on a daily basis.

Preoperatively, we were able to assess the factors that we sought to measure in the intervention group only because the patients in the control group were admitted to the department where the surgery was performed on the day of the surgery or in the afternoon immediately before the surgery. In contrast, in the case of the intervention group, there was ample time to collect the data during the prehabilitation program. Thus, in our manuscript, were compared only those data that were collected at the same time in the case of both groups. Multivariate models were not performed in the study because the database did not contain data identified as confounders.

## 6. Conclusions

Our research results confirmed that, as a result of a prehabilitation program before complete hip and knee replacement surgery, the length of the postoperative hospital stay decreased significantly, which significantly affected the decrease in the patient’s care costs. Our study confirmed that prehabilitation effectively aids in the restoration of functional abilities in the postoperative period. We found a significant improvement in the flexion motion range of the hip immediately after surgery in the intervention group. At the end of the postoperative period, we also found a significant increase in the flexion motion of the knee in the intervention group. The results based on the quality of life scale suggest that we achieved significantly better results in the postoperative period in the case of both intervention groups.

We consider it extremely important that the program be continued and expanded and recommend that the prehabilitation program be included in the tasks related to the knee and hip prosthesis. The applied patient education further improved the effectiveness of our program. It was useful for the patient to be informed about every step of the process, enabling them to contribute to their quick recovery.

## Figures and Tables

**Figure 1 ijerph-19-15067-f001:**
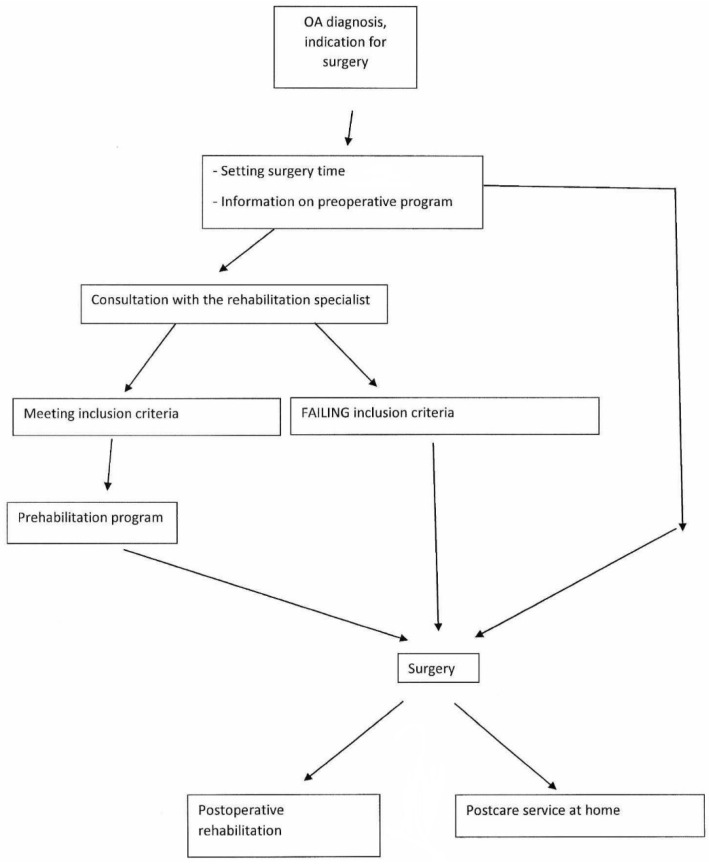
Flow chart showing the recruitment of patients.

**Figure 2 ijerph-19-15067-f002:**
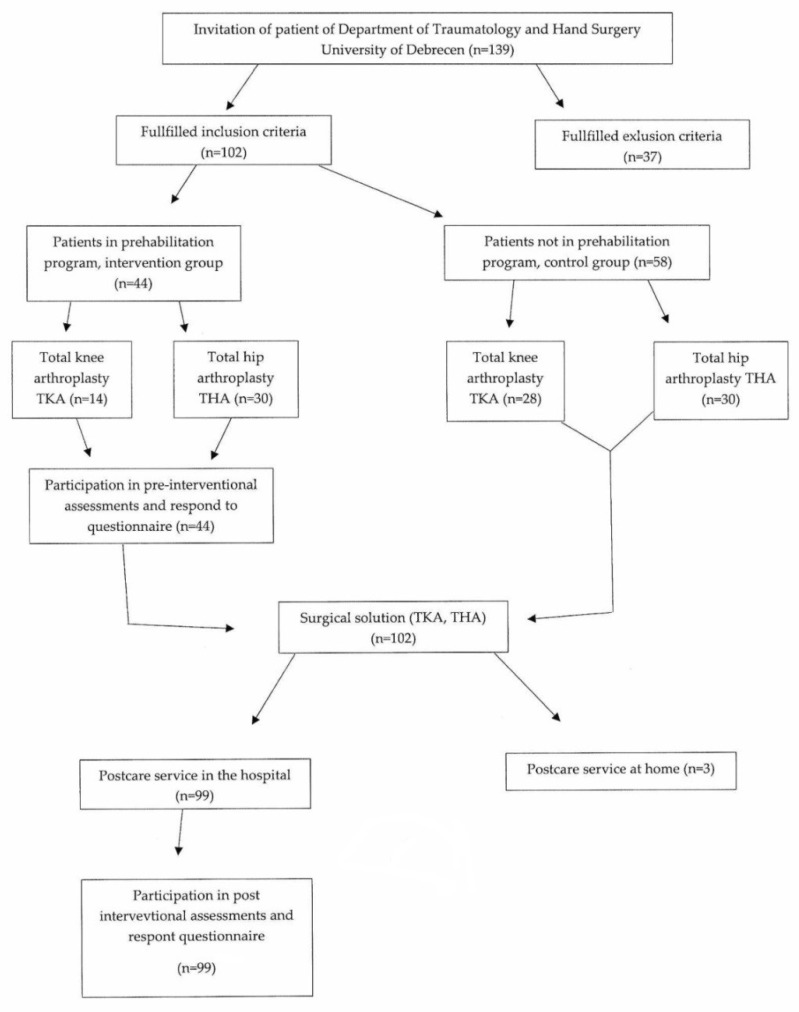
Flow chart of the novel patient pathway.

**Table 1 ijerph-19-15067-t001:** Demographic data. Gender (female/male), age (mean ± SD), and *p*-value are shown for different groups.

Surgery	Gender	Group	Number	Mean	±SD	*p*-Value (Age Comparison)	*p*-Value (Gender Distribution)
Total knee arthroplasty	Female	Control	19	69.10	7.99	0.897	0.719
		Intervention	11	69.18	6.44		
	Male	Control	9	71.44	6.54	0.064	
		Intervention	3	80.00	2.00		
Total hip arthroplasty	Female	Control	22	73.04	8.55	0.406	0.278
		Intervention	16	70.75	9.88		
	Male	Control	8	67.75	9.73	0.042	
		Intervention	11	57.90	13.38		

**Table 2 ijerph-19-15067-t002:** Range of motion of the hip and knee joints of endoprosthesis patients. Data on the flexion, extension after hip and knee surgery, and abduction after hip surgery are shown for the control and intervention groups, respectively. ns. = non-significant result.

			After Surgery		Leaving Hospital				
			Median	IQR	Median	IQR	*p*-Value (After Surgery vs. Leaving Hospital)	*p*-Value (After Surgery Control vs. Intervention)	*p*-Value (Leaving Hospital Control vs. Intervention)
Flexion (°)	Hip surgery	Control group	60	50–70	90	81.25–90	<0.001		
		Intervention group	75	70–80	90	85–90	<0.001	<0.01	ns.
	Knee surgery	Control group	65	45–70	90	90–90	<0.001		
		Intervention group	57.5	51.25–65	100	95–100	<0.001	ns.	<0.01
Extension (°)	Hip surgery	Control group	0	0–5	0	0–3	<0.05		
		Intervention group	0	0–4	0	0–1.5	ns.	ns.	ns.
	Knee surgery	Control group	2.5	0–6.25	0	0–3.5	<0.01		
		Intervention group	3	0–5	0	0–0	<0.01	ns.	ns.
Abduction (°)	Hip surgery	Control group	15	10–20	25	20–30	<0.001		
		Intervention group	15	10–20	28	24.5–30	<0.001	ns.	ns.

**Table 3 ijerph-19-15067-t003:** Oxford Hip and Knee Score among hip and knee endoprosthesis patients. Data after hip and knee surgery are shown for the control and intervention groups, respectively.

		After Surgery		
		Median	IQR	*p*-Value (Control vs. Intervention)
Hip surgery (points)	Control group	25	25–30	
	Intervention group	33	31.5–35	˂0.001
Knee surgery (points)	Control group	30.5	30–35	
	Intervention group	35	33–35	˂0.05

## Data Availability

Data are available from the authors at reasonable written request after authorization by the Data Protection Office of the University of Debrecen, Hungary.

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
