# Peer review of "Cost-Effective Healthcare in Rehabilitation: Physiotherapy for Total Endoprosthesis Surgeries from Prehabilitation to Function Restoration"

_ijerph, 2022, doi:10.3390/ijerph192215067_

Round 1
Reviewer 1 Report (New Reviewer)
Dear authors:
First of all I would like to congratulate you on your research work. Then I would like to suggest some changes to improve your manuscript:
- At the beginning you talk about incidence and prevalence, but it is not clear whether you are talking about worldwide, in Europe, in Debrecen? Please clarify.
- In the methods section, you comment that you do not make a previous study of the sample, this is a big bias in your work. In my opinion, you should modify this section.
- In addition to talking about exclusion criteria, you should make clear the inclusion criteria.
- They also talk about an exercise programme, electrotherapy and massage, but they should explain in depth what type of exercises (eccentric, isometric, isotonic), in what joint range, repetitions..... In addition, they should explain what kind of electrotherapy is applied and for what purpose it is applied. Finally, they should clarify what kind of manual therapy is applied, on which muscles and for what purpose.... they should improve this section a lot.
- They should include a flowchart on the recruitment of COnsort patients.
- The section on the ethics committee should be expanded, by which committee it was approved, date, follow-up number. In addition, they should keep a record of the trial in clinicaltrial.
- The discussion section is very brief, they should expand it and comment on more similar studies or studies that observe the same variables in order to increase the number of bibliographic citations.
In general, the article is good, but certain aspects need to be improved in order to be published.
Best regards.
Author Response
Response to Reviewer 1 Comments
Dear Reviewer! Thank you for your careful review, for your important comments and suggestions and for giving us the opportunity to resubmit a revised version of the manuscript “Cost-effective healthcare in rehabilitation: physiotherapy in total endoprosthesis surgeries from prehabilitation to function restoration” for publication in the Special Issue of International Journal of Environmental Research and Public Health
Point 1: At the beginning you talk about incidence and prevalence, but it is not clear whether you are talking about worldwide, in Europe, in Debrecen? Please clarify.
Response 1: We have made the necessary changes in the Introduction as requested.
Point 2: In the methods section, you comment that you do not make a previous study of the sample, this is a big bias in your work. In my opinion, you should modify this section.
Response 2: The estimated sample was calculated using the "sampsi" command of Intercooled Stata v13. The type I error was set to 5% and the statistical power was set to 90%. Refusal rate was also taken into consideration. The calculation was based on relevant article.
Reference:
Calatayud J, Casaña J, Ezzatvar Y, Jakobsen MD, Sundstrup E, Andersen LL. High-intensity preoperative training improves physical and functional recovery in the early post-operative periods after total knee arthroplasty: a randomized controlled trial. Knee Surg Sports Traumatol Arthrosc. 2017 Sep;25(9):2864-2872. doi: 10.1007/s00167-016-3985-5. Epub 2016 Jan 14. PMID: 26768606.
Point 3: In addition to talking about exclusion criteria, you should make clear the inclusion criteria.
Response 3:
We have made all the necessary changes in the Methods section as requested.
Answers to the questions:
Inclusion criteria for intervention groups were:
-Ability to participate in the prehabilitation program on a daily basis and carry out the gymnastics programme set up by the team for 30 minutes a day.
-Willingness to cooperate with the professionals involved in the prehabilitation programme, and accept and follow instructions.
Point 4: They also talk about an exercise programme, electrotherapy and massage, but they should explain in depth what type of exercises (eccentric, isometric, isotonic), in what joint range, repetitions..... In addition, they should explain what kind of electrotherapy is applied and for what purpose it is applied. Finally, they should clarify what kind of manual therapy is applied, on which muscles and for what purpose.... they should improve this section a lot.
Response 4: We have made all the necessary changes in the Methods section as requested.
Answers to the questions:
Intervention
During the prehabilitation program for hip and knee joints daily exercise lasted for 30 minutes. It consisted of 4 parts:
Introduction: Patients were in a relaxed state, resulting in reduced muscle tension caused by pain. We increased circulation in the lower extremities using leg presses and ankle circles. We asked patients to perform breathing exercises, mobilizing the upper extremities and the chest. The aim of these exercises was to prepare the cardio-vascular system for physical load, to support tissue oxygenization and to achieve psychological stress relief.
- Increasing range of motion in the affected hip or knee joint:
Hip joint:
- At first, improvement of flexion was performed by patients in the supine position and passive/assisted active movements guided by a physiotherapist, starting with short lever arm followed later by active movements also with short as well as long lever arm. In the lateral position we had patients practise active flexion movements only.
- Abduction and adduction movements were carried out by patients in the supine and lateral positions. At first abduction was performed against gravity and then, to increase resistance, smaller ankle weights /0.5 kg) or flexible rubber bands were used to make the agonist-antagonist reflex muscle relief effect more effective.
- Extension movements were performed in the supine position using isometric exercises, mainly through activation of the gluteal muscles, and in the lateral position, through active movements, and finally, for a short period of time ( 5-10 min max) with the patients positioned in the prone position.
- During practising plane movements, great emphasis was placed on keeping the corrected rotation middle position.
- During increase of the range of flexion–extension movement all three positions were used (supine, prone, and lateral), using only active relaxation and stretching techniques (post isometric relaxation; reciprocal innervation; contract-relax pnf stretching). In case of extension particular attention was paid to practising while keeping full range of motion.
III. Increasing muscle strength:
Here emphasis was on strengthening the extensor and abductor muscles as well as m. quadriceps. In case of the hip joint, our aim was to correct the muscle imbalance caused by typical flexion and abduction contracture. To this end, the most frequently used positions were the classical horizontal positons (supine, prone, lateral, and all fours) and the alternative variants of these, e.g. when the patient is lying on the therapy bed in the prone position hip height and the two lower extremities are positioned on the floor, hanging from the end of the bed. In this position concentric and excentric movement of the hip extensors can be easily performed and the effectiveness of the training can be further increased using less resistance. In addition, if the patients’ general condition made this possible, they performed muscle strengthening exercises using wall bars. During these exercises, concentric and excentric strengthening of the abductors performed in a corrected position was mainly practised using different sizes of resistance.
A further aim of our study was to strengthen the quadriceps muscles, which are characterised by considerable weakening in degenerative diseases of both the hip and the knee joints. Typically, strengthening was carried out with patients in the supine position, in the affected leg, using extended hip joint in starting position, working with short and long lever arm. We increased load gradually by increasing the number of repetitions of the exercises, using gravity only or in some cases weaker rubber bands as resistance. In order to avoid paracoordination or compensatory movements, the pelvis was always stabilized by positioning the other lower leg, or, if necessary, manual fixing by the physiotherapist.
The intensity and strength of the exercises, as well as the number of repetitions were determined in every case by the current condition of the patient; we made sure we were not causing or increasing pain during the movements.
The tools used during the exercise program helped maintain patients’ interest and made the exercises more varied. Soft Balls, rubber bands of different strengths, Dynair cushions, and Fit Balls were used.
- Cooldown:
The last stage of the exercise program featured gradual load decrease as well as relaxing exercises, active stretching and breathing exercises to restore circulation.
During the prehabilitation program members of the intervention group also received electrotherapy and therapist massage in addition to the exercises. Electrotherapy treatments were meant to reduce pain in the given area, improve circulation and activate and strengthen the muscles around the joints. To this end, interferential therapy, the application of selective stimulation, and electromagnetic therapies were also incorporated in the program.
In using classical therapist massage we intended to increase range of motion of the joint, improve circulation and relaxing painful, adhesions.
Point 5: They should include a flowchart on the recruitment of COnsort patients.
Response 5: We have made the necessary changes in the Methods section as requested.
Flowchart:
Point 6: The section on the ethics committee should be expanded, by which committee it was approved, date, follow-up number. In addition, they should keep a record of the trial in clinicaltrial.
Response 6: We have made all the necessary changes in the Methods section as requested.
Answers to the questions:
The study was approved by the Research Ethics Committee of the Kenézy Gyula Hospital of the University of Debrecen (É/2020/01/23), and participants gave informed consent to the collection and processing of data.
Point 7: The discussion section is very brief, they should expand it and comment on more similar studies or studies that observe the same variables in order to increase the number of bibliographic citations.
Response 7: We have made the necessary changes in the Discussion section as requested.
Discussion
Our research investigated the efficiency of a novel and special patient pathway in the course of which a patient receives a recommendation for participation in prehabilitation from the physician who is going to operate them. During the prehabilitation program the patient meets the members of the intervention team including those specialists with whom they are going to work together in the postrehabilitation period. This option increases the patient’s trust in the successful outcome of the surgery.
Our results showed that patients in the prehabilitation program were able to leave the hospital 7.5 days earlier on average in the postoperative rehabilitation process compared to patients in the control group. In addition, the results of the range-of-motion measurements showed that the intervention group was able to achieve better functional abilities during postoperative rehabilitation despite shorter hospital stays. These results were in accordance with those reported by Oosting E. et al., who found a significant difference between prehabilitation and control patients undergoing hip endoprosthesis surgery in the Timed Up & Go test (2.9 s; 95% confidence interval [CI], 0.9-6.6) and the 6-minute walk test (41m; 95% CI, 8-74) [8]. In their systematic review and meta-analysis Moyer et al. also found that preoperative exercise and education for patients undergoing total hip and knee arthroplasty significantly improved the postoperative function in both patient groups (THA, TKA) with similar improvement. Furthermore, in their review length of hospital stay was significantly shorter after TKA and THA (23). Dlott et al. also demonstrated that elective primary total hip and knee arthroplasty patient education delivered by nurses before elective total hip and knee arthroplasty reduced the average length of hospital stay, postoperative emergency department (ED) visits, and subsequently increased the use of telerehabilitation. (24)
In their review, Widmer et al. compared the two forms of prehabilitation: exercise therapy and patient education in patients with total hip arthroplasty (THA). (25) They concluded that, in the postoperative period, exercise-based prehabilitation patients had better results in the functional tests compared with patients with no prehabilitation. In contrast, patient education alone did not significantly improve patients’ results. (25,26).
Several previous studies and summary reports have examined the effectiveness of prehabilitation in patients with hip and knee replacement. (26,28) However, results are ambiguous, a fact possibly explained by differences in the types of interventions used during prehabilitation and the lengths of the prehabilitation period. The importance of the time factor was also highlighted by Widmer et al. The effectiveness of prehabilitation shows a close relationship with the length of the intervention and the intensity of the therapies; in other words, the effectiveness of prehabilitation shows dose-dependence both in outpatiens and inpatients. (25) The question arises whether the effectiveness of prehabilitation is affected by socio-demographic factors (e.g. age, gender). However, as of now, the role of these factors has not been confirmed in patients with hip replacement. (25)
In our study data on quality of life scales in the intervention group showed a very significant difference for hip prosthesis surgeries, and patients receiving knee prosthesis also reported a significantly better quality of life. Quality of life scale data showed highly significant differences in the intervention group for hip replacement surgery, and patients who underwent knee replacement also reported significantly better quality of life.
Due to shorter hospital stays, the cost of hospital care was reduced, resulting in financial savings for the healthcare system. In our research postoperative costs of hip and knee endoprosthesis implantation decreased by 26% and 16%, respectively. Our results are comparable to those in previous studies, such as those by Jones et al. and Huang et al., who reported that patients in the study spent fewer days in hospital and thus had more favourable hospital costs (mean: 7.12 days; p = 0.027 and mean: 123.73, which is 427.4 USD or 302.1 EUR, p = 0.001) compared to the control group (2,13).
The efficiency of our prehabilitation program was demonstrated by the fact that three of our patients with hip replacement surgery did not need postoperative rehabilitation in hospital at all due to their favourable functional condition. Butler et al. and Konnyu et al. have demonstrated that THA patients who have undergone prehabilitation require shorter rehabilitation, as they reach adequate functionality and can leave the hospital earlier. (27,28)
The results obtained in our study support the usefulness of the new patient pathway, and our results are in many respects consistent with previous reports. In their studies, Rajrishi Sharma et al found a significant result in hospital stay in favour of prehabilitation (11). The results of studies by Pascale Gränicher et al. also suggest that preoperative therapy influences movements in a positive direction (12). In the conclusion of their article, Plenge et al. described the importance of developing a perioperative multidisciplinary patient care protocol for hip and knee arthroplasty patients based on the results of preoperative intervention. (29)

Reviewer 2 Report (Previous Reviewer 2)
This study supported preoperative rehabilitation for THA or TKA effectively control postoperative medical cost. Two issues to be clearly described;
1. Method of allocation: random or not?
2. start point of rehabilitation: postoperative rehab started at day 0, or 1?
Corrections;
P9, L25: typically 60-----years-old ?
P10, Ref 5: J Bone Joint Surg Am
Author Response
Response to Reviewer 2 Comments
Dear Reviewer! Thank you for your careful review, for your important comments and suggestions and for giving us the opportunity to resubmit a revised version of the manuscript “Cost-effective healthcare in rehabilitation: physiotherapy in total endoprosthesis surgeries from prehabilitation to function restoration” for publication in the Special Issue of International Journal of Environmental Research and Public Health
Point 1: Method of allocation: random or not?
Response 1: Participants were randomly allocated into the intervention and control groups.
Point 2: Start point of rehabilitation: postoperative rehab started at day 0, or 1?
Response 2: We have made all the necessary changes in the Methods section as requested.
Answers to the question:
On the 8th to 10th day after surgery, following suture removal, patients are transferred to the rehabilitation ward for a medical consultation and physiotherapy assessment on the same day, with post-operative rehabilitation starting on the first day.
Point 3: Corrections;
P9, L25: typically 60-----years-old ?
P10, Ref 5: J Bone Joint Surg Am
Response 3:
We have made all the necessary changes in the manuscript as requested.

This manuscript is a resubmission of an earlier submission. The following is a list of the peer review reports and author responses from that submission.
Round 1
Reviewer 1 Report
The manuscript represents research regarding the effects of prehabilitation on the outcomes of TKA and THA. A lot research has been done in that field. These are my comments and suggestions:
Abstract:
Third sentence of the abstract seems unfinished. It is the same with the fifth sentence. Abstract requires proofreading and editing. Aim of the research is not clearly stated in the abstract. More quantitative data should be included in the abstract.
Introduction:
Besides physiotherapy, there are also other possible treatments for OA (pharmacological therapy). Aim of the research should be stated more clearly. Description of the prehabilitation programme should be placed in the Methods part of the manuscript. In general, Introduction part is too long.
Methods:
Methods should be written according to standard reporting guidelines. Furthermore, past tense is used when describing methods. How did you justify your sample size? It is quite small with very unequal number of participants in two groups. Give more details regarding measurements of the outcomes. How did you ensure that they were valid and reliable? Give more details regarding the surgical procedure and post-surgery care.
Results:
Is it possible that the hospital stay was 20+ days? Does it include postsurgical rehabilitation? This is quite long. If you used non-parametric test then it is customary to present median and IQR values. Did you make baseline (pre-surgery) measurements? This would allow to see changes pre- and post and methodological quality of the study would be higher.
Discussion:
Why is your research important and novel? This should be at the start of the Discussion. Please, rewrite and include all study limitations.
Conclusion:
Should be rewritten and summarize your results.
Author Response
Response to Reviewer 1 Comments
Dear Reviewer! Thank you for your careful review, for your important comments and suggestions and for giving us the opportunity to resubmit a revised version of the manuscript “Cost-effective healthcare in rehabilitation: physiotherapy in total endoprosthesis surgeries from prehabilitation to function restoration” for publication in the Special Issue of International Journal of Environmental Research and Public Health
Point 1: Abstract:
Third sentence of the abstract seems unfinished. It is the same with the fifth sentence. Abstract requires proofreading and editing. Aim of the research is not clearly stated in the abstract. More quantitative data should be included in the abstract.
Response 1: We have made all the necessary changes in the Abstract as requested.
Point 2: Introduction:
Besides physiotherapy, there are also other possible treatments for OA (pharmacological therapy). Aim of the research should be stated more clearly. Description of the prehabilitation programme should be placed in the Methods part of the manuscript. In general, Introduction part is too long.
Response 2: We have made all the necessary changes in the Introduction as requested.
Point 3:Methods:
Methods should be written according to standard reporting guidelines. Furthermore, past tense is used when describing methods. How did you justify your sample size? It is quite small with very unequal number of participants in two groups. Give more details regarding measurements of the outcomes. How did you ensure that they were valid and reliable? Give more details regarding the surgical procedure and post-surgery care.
Response 3: We have made all the necessary changes in the Materials and Methods section as requested.
Answers to the questions:
The size of the intervention group was affected by the fact that many of the patients that could have been included in the program were not because they failed the inclusion criteria. The number of participants in the study was further limited by the fact that both the physicians and the patients were worried that the prehabilitaion happening in the hospital might increase the risk of nosocomial infections, which could lead to the postponement of the planned surgery. In our insitution patients with prostheses are usually over 60; these people are specially affected by their being away from their families, hence the chances of them uncdertaking even more time to be spent in hospital postperatively were slim. When we determined sample size for our study, we considered earlier publications, too, where the effciciency of prehabiliation and postoperative treatments in patients with hip and knee arthroplasty had been examined on similar or smaller numbers of cases. Maarit et al examined the introduction of new exercises during late rehabilitation of full knee replacement in a group of seven patients as a result of which postoperative movements did not lead to an increase in stress and pain levels. (1)
In their study Joaguin et al examined the efficiency of preoperative training in OA patients with a view to improve early postoperative results after surgery. In their study the total number of patients was 44 (control group: 22, and intervention group: 22 ) due to a significant drop-out rate and failure to meet inclusion criteria. (2)
From a methodology spoint of view the same number of patients in the control and the intervention group would be ideal but it is not compulsory because the results of the staitistical results are not significantly affected by the difference between the numbers of participants.
The Oxford Hip and Knee Score is an internationally recognized, validated quality of life scale which is extensively used. (3,4)
A goniometer is extensively used in the field of physiotherapy to measure range of motion at joints. (5)
Point 4: Results:
Is it possible that the hospital stay was 20+ days? Does it include postsurgical rehabilitation? This is quite long. If you used non-parametric test then it is customary to present median and IQR values. Did you make baseline (pre-surgery) measurements? This would allow to see changes pre- and post and methodological quality of the study would be higher.
Response 4: We have made all the necessary changes in the Results section as requested. Responses to the questions:
In our institution patients usually spend 8 days after surgery in the surgery department and are then transfered to the Rehabilitation Ward itself. By international comparison, unfortunately, patients do spend longer in the Rehabilitaion Ward. This has several reasons. In Hungary home care is poorly supported as a result of which care providers cannot employ sufficent numbers of rehabilitation professionals. This results in long waiting lists, which, in turn leads to delays in the start of therapy. Thus patients find it safer to spend the rehabilitaion period in hospital, enabling them to use the services of a professional staff (nurses, physiotherapists and physicians) 24 hours a day. Another argument for rehabilitation as part of inpatient care in hospital is that a significant portion of patients would find it difficult to access outpatient care due to their negative socio-economic postion (income, education, place of residence, subjective welfare) immediately after surgery. All Hungarian citizens covered by social security are legally entitled to this form of care.(6)
Preoperatively we were able to assess the factors that we wanted to measure in the intervention group only because the patients in the control group were admitted to the department where the surgery was performed only on the day of the surgery or the afternoon immediately before the surgery. In contrast, in the case of the intervention group, there was ample time to collect data during the prehabilitation program. Thus in our manuscript were compared only those data that were collected at the same time in both groups.
It it had been possible, we would have collected all data before the surgery as well in both groups. We agree with the reviewer that this would have enhanced the quality of our study.
Point 5: Discussion:
Why is your research important and novel? This should be at the start of the Discussion. Please, rewrite and include all study limitations.
Response 5: We have made all the necessary changes in the Discussion section as requested.
Point 6: Conclusion:
Should be rewritten and summarize your results.
Response 6: We have made all the necessary changes in the Conclusion as requested.
References
- Maarit J., Antti L., † Simon W, Taavi P, Niina K, Sulin C, Juha P, Konsta P, Mika L, Raija K, Timo J, Ari H, Eeva A, Movement characteristics during customized exergames after total knee replacement in older adults Published online 2022 Jul 27. doi: 10.3389/fspor.2022.91521 PMCID: PMC9363837 PMID: 35966111
- Joaquin C, Jose C, Yasmin E, Markus D. J, Emil S & Lars L. A High-intensity preoperative training improves physical and functional recovery in the early post-operative periods after total knee arthroplasty: a randomized controlled trial Knee Surgery, Sports Traumatology, Arthroscopy volume 25, pages2864–2872 (2017) DOI: 10.1007/s00167-016-3985-5 PMID: 26768606
- Dawson, R. Fitzpatrick, D. Murray, A. Carr Questionnaire on the perceptions of patients about total knee replacement The Journal of Bone and Joint Surgery. British volumeVol. 80-B, No. 1 Published Online:1 Jan 1998https://doi.org/10.1302/0301-620X.80B1.0800063
- Shiraz A. Sabah,Ruth Knight,Abtin Alvand,David J. Beard,Andrew J. Price Early patient-reported outcomes from primary hip and knee arthroplasty have improved over the past seven years an analysis of the NHS PROMs dataset The Bone & Joint JournalVol.104-B,No.6Published Online:31 May 2022https://doi.org/10.1302/0301-620X.104B6.BJJ-2021-1577.R1
- Holm I, Bolstad B, Lütken T, Ervik A, Røkkum M, Steen H. Reliability of goniometric measurements and visual estimates of hip ROM in patients with osteoarthrosis. Physiother Res Int. 2000;5(4):241-8. doi: 10.1002/pri.204. PMID: 11129666.
- Tasks of the National Health Insurance Fund of Hungary (Hungarian acronym: NEAK) http://www.neak.gov.hu/felso_menu/lakossagnak/ellatas_magyarorszagon/egeszsegugyi_ellatasok/otthoni_szakapolas_hospice/otthoni_szakapolas_hospice.html
requested 04/10/2022

Reviewer 2 Report
This study reported preoperative rehabilitation had a functionally and financially superiority for THA and TKA surgeries by prospective case control study with no random allocation. There are several points to improve the manuscript.
1. Redundant sentences: three times appear
P2,L16~L19; P7,L19~L20; P7,L39~P8,L5
2. To clarify financially superiority in intervention group, preoperative rehab cost must be included.
3. To clarify superiority in preoperative rehabilitation, postoperative rehab must be same in both groups. If this study compared old clinical path and new clinical path, this study design can be significant.
4. Corrections
P2,L35: Ran-----ran
P5,L3: GONIOMETER-----goniometer
P7,L13: left v.?
Author Response
Response to Reviewer 2 Comments
Dear Reviewer! Thank you for your careful review, for your important comments and suggestions and for giving us the opportunity to resubmit a revised version of the manuscript “Cost-effective healthcare in rehabilitation: physiotherapy in total endoprosthesis surgeries from prehabilitation to function restoration” for publication in the Special Issue of International Journal of Environmental Research and Public Health
Point 1: Redundant sentences: three times appear
P2,L16~L19; P7,L19~L20; P7,L39~P8,L5
Response 1: We have made all the changes in the manuscript as requested.
Point 2: To clarify financially superiority in intervention group, preoperative rehab cost must be included.
Response 2: The aim of our study was to assess the effect of prehabiliation on the postoperative period. In the Discussion we stated (Limitations) that the full cost of all treatments grew as a result of prehabilitation expenses, which explains why we were unable to demonstrate cost savings in terms of full (pre- and postoperative) hospital costs. As we pointed out in our study, the costs of the prehabilitation period can be reduced if patients use the outpatient service or that of the day hospital.
Point 3: To clarify superiority in preoperative rehabilitation, postoperative rehab must be same in both groups. If this study compared old clinical path and new clinical path, this study design can be significant.
Response 3: Postoperative rehabilitation took place in the same way in both patient groups (knee and hip arthroplasty) as well as in the control and the intervention group. The difference in the control and intervention group was in participation in the preoperative program.
Point 4: Corrections
P2,L35: Ran-----ran
P5,L3: GONIOMETER-----goniometer
P7,L13: left v.?
Response 4: We have made all the necessary changes in the manuscript.

Reviewer 3 Report
Dear authors,
It was my pleasure to assess your manuscript. It is an interesting manuscript, congratulation for the results and for selecting this subject. After assessing it the following issues raised my concern or represent suggestions that in my opinion could increase the quality of your paper:
- Abstract:
o “Data collection about the effects of prehabilitation regarding the length and cost of postoperative rehabilitation in hospital, and the restoration of functional abilities. Case control study was performed.” If it is possible, I suggest you to rewrite the structure in an more academic manner.
o “and complex physiotherapy after endoprosthesis surgery” An action verb is missing. I think.
- Introduction:
o You are using both “osteoarthritis” and “Osteoarthrosis” – please use only one word.
o “functional restoration after arthroplasty [7]” An “.” is missing at the end of the sentence.
- Results:
o You report the following hospitalization period for intervention vs. control group “(25.89 ±6.03 days, n = 27 vs. 35.33 ±12.93 days, n = 30).” Why a so long period of hospitalization even for the intervention group? Because the age of the patients doesn’t seem to be so increased, I supposes that this are cases that didn’t had complications after the surgery. Just a simple search on TKA hospitalization period brings plenty of studies, for e.g Denaro et al J. Clin. Med. 2022, 11(8), 2114; https://doi.org/10.3390/jcm11082114 report a median length of hospital stay was 3 (IQR 3, 4) days. Are this cemented of uncemented THA? What approaches are used for TKA and THA? When do you start the postop rehab program? Weight bearing immediately after surgery with aids? What aids do you use and for what period of time? You provide some explanations in the limitations section of the manuscript.
- Limitations:
o “The total cost per patient in prehabilitation increased significantly by the end of the program because the majority of patients chose to stay in hospital for the prehabilitation program.” Dose the health care system from Hungary supports this increased hospitalization costs and periods? Patients are required to pay for the surgeries via private healthcare insurance?
o I think that it would be cheaper for the healthcare system if the patients will learn the exercise during the first days after surgery, to be discharged from the hospital and to be seen after three weeks from the surgery in an outpatient clinic and to be reevaluated?
Author Response
Response to Reviewer 3 Comments
Dear Reviewer! Thank you for your careful review, for your important comments and suggestions and for giving us the opportunity to resubmit a revised version of the manuscript “Cost-effective healthcare in rehabilitation: physiotherapy in total endoprosthesis surgeries from prehabilitation to function restoration” for publication in the Special Issue of International Journal of Environmental Research and Public Health
Point 1: Abstract:
o “Data collection about the effects of prehabilitation regarding the length and cost of postoperative rehabilitation in hospital, and the restoration of functional abilities. Case control study was performed.” If it is possible, I suggest you to rewrite the structure in an more academic manner.
o “and complex physiotherapy after endoprosthesis surgery” An action verb is missing. I think.
- Introduction:
o You are using both “osteoarthritis” and “Osteoarthrosis” – please use only one word.
o “functional restoration after arthroplasty [7]” An “.” is missing at the end of the sentence.
Response 1: We have made all the changes in the manuscript as requested.
Point 2: Results:
You report the following hospitalization period for intervention vs. control group “(25.89 ±6.03 days, n = 27 vs. 35.33 ±12.93 days, n = 30).” Why a so long period of hospitalization even for the intervention group? Because the age of the patients doesn’t seem to be so increased, I supposes that this are cases that didn’t had complications after the surgery. Just a simple search on TKA hospitalization period brings plenty of studies, for e.g Denaro et al J. Clin. Med. 2022, 11(8), 2114; https://doi.org/10.3390/jcm11082114 report a median length of hospital stay was 3 (IQR 3, 4) days. Are this cemented of uncemented THA? What approaches are used for TKA and THA? When do you start the postop rehab program? Weight bearing immediately after surgery with aids? What aids do you use and for what period of time? You provide some explanations in the limitations section of the manuscript.
Response 2:
In our institution patients usually spend 8 days after surgery in the surgery department and are then transfered to the Rehabilitation Ward itself. By international comparison, unfortunately, patients do spend longer in the Rehabilitaion Ward. This has several reasons. In Hungary home care is poorly supported as a result of which care providers cannot employ sufficent numbers of rehabilitation professionals. This results in long waiting lists, which, in turn leads to delays in the start of therapy. Thus patients find it safer to spend the rehabilitaion period in hospital, enabling them to use the services of a professional staff (nurses, physiotherapists and physicians) 24 hours a day. Another argument for rehabilitation as part of inpatient care in hospital is that a significant portion of patients would find it difficult to access outpatient care due to their negative socio-economic postion (income, education, place of residence, subjective welfare) immediately after surgery. All Hungarian citizens covered by social security are legally entitled to this form of care.(1)
The total number of patients participating in our research was 99; 57 underwent total hip and 42 had knee surgery. Of these 62 patients had cement fixation and 37 did not. This is important from the aspect of load as in the case of cement fixation the prosthesis is fixed to the bone with two-component bone cement and load can be started on the first day after surgery With uncemented fixation the bone grows around the prosthesis, allowing partial load on the extremity on the first day after surgery using a walking frame to be continued for 4 weeks, followed by the use of crutches. Use of crutches can be completely stopped in week 6.
Point 3: Limitations:
o “The total cost per patient in prehabilitation increased significantly by the end of the program because the majority of patients chose to stay in hospital for the prehabilitation program.” Dose the health care system from Hungary supports this increased hospitalization costs and periods? Patients are required to pay for the surgeries via private healthcare insurance?
o I think that it would be cheaper for the healthcare system if the patients will learn the exercise during the first days after surgery, to be discharged from the hospital and to be seen after three weeks from the surgery in an outpatient clinic and to be reevaluated?
Response 3: As we pointed out in our response 2, all citizens covered by social security are legally entitled to this form of care, and, due to their negative socio-economic position (income, education, place of residence, subjective welfare), many patients would find it difficult to go to daily outpatient care every day.
Hivatkozás
- Tasks of the National Health Insurance Fund of Hungary (Hungarian acronym: NEAK) http://www.neak.gov.hu/felso_menu/lakossagnak/ellatas_magyarorszagon/egeszsegugyi_ellatasok/otthoni_szakapolas_hospice/otthoni_szakapolas_hospice.html
requested 04/10/2022

Round 2
Reviewer 1 Report
I am satisfied with the improvements of the article.
Reviewer 3 Report
Dear authors,The performed changes definitely improved the quality of the manuscript.